# Fecal Microbiota and Diet Composition of Buryatian Horses Grazing Warm- and Cold-Season Grass Pastures

**DOI:** 10.3390/microorganisms11081947

**Published:** 2023-07-30

**Authors:** Svetlana Zaitseva, Olga Dagurova, Aryuna Radnagurueva, Aleksandra Kozlova, Anna Izotova, Anastasia Krylova, Sergey Noskov, Shahjahon Begmatov, Ekaterina Patutina, Darima D. Barkhutova

**Affiliations:** 1Institute of General and Experimental Biology SD RAS, Sakhyanovoy str., 6, 670047 Ulan-Ude, Russia; dagur-ol@mail.ru (O.D.); aryuna_rg@mail.ru (A.R.); darima_bar@mail.ru (D.D.B.); 2Kurchatov Center for Genome Research, NRC.urchatov Institute, 123182 Moscow, Russia; amiandwho@gmail.com (A.K.); anu.izotova@mail.ru (A.I.); krylova.gen@gmail.com (A.K.); sergey.noskov.2001@icloud.com (S.N.); 3Institute of Bioengineering, Research Center of Biotechnology of the Russian Academy of Sciences, Leninsky Prosp, bld. 33-2, 119071 Moscow, Russia; shabegmatov@gmail.com (S.B.); e.o.patutina@gmail.com (E.P.)

**Keywords:** microbial diversity, 16S rRNA, horse indigenous breed

## Abstract

The Buryatian horse is an ancient breed and, as an indigenous breed, they have unique adaptive abilities to use scarce pastures, graze in winter, and survive in harsh conditions with minimal human care. In this study, fecal microbiota of Buryatian horses grazing in the warm and cold seasons were investigated using NGS technology on the Illumina MiSeq platform. We hypothesized that the composition of microbial communities in the feces of horses maintained on pasture would change in the different seasons, depending on the grass availability and different plant diets. We conducted microhistological fecal studies of horse diet composition on steppe pasture. The alpha diversity analysis showed horses had a more abundant and diverse gut microbiota in summer. There were significant effects on the beta diversity of microbial families, which were clustered by the warm and cold season in a principal coordinate analysis (PCoA), with 45% of the variation explained by two principal coordinates. This clustering by season was further confirmed by the significant differences observed in the relative abundances of microbial families and genera. The obtained results can serve as an experimental substantiation for further study of the impact of pasture grasses, which have a pharmacological effect, on the diversity of the gut microbiome and horse health.

## 1. Introduction

The horse gut microbial community plays a decisive role in maintaining intestinal and organism health [1,2,3]. Several research groups have studied the effects of different factors such as age, breed, environmental conditions (such as season, diet, pasture access, fasting, transportation, and exercise), and the use of pre- and probiotics treatment on the structure of these microbial communities [2,3,4,5,6,7]. It is thought that the ancient horse microbiota was richer and more resilient, conferring protection against environmental challenges, but modern veterinary practices and lifestyles may have changed it [8]. Thus, the comparison of the fecal microbiome from indigenous horses with an ancestral-like lifestyle from different geographical locations enables the characterization of the core horse microbiome and an understanding of the impact of local diets and conditions [1].

The conservation of native domestic animal breeds is among the most important indicators of sustainable agricultural development [9]. Indigenous breeds contribute greatly to the genetic diversity of the world’s domesticated animal populations and have high significance for socio-economic and cultural heritage [10,11]. Indigenous breeds of horses have unique adaptive qualities and are able to use scarce pastures, graze in winter, and survive in harsh conditions with minimal human care [12]. Local environmental conditions, nature, and the nomadic form of husbandry were the prime factors determining the evolution of aboriginal horse breeds [13].

The Buryatian horse is one of the most ancient breeds that are actively used in agriculture today [13]. The Buryatian breed of horse is common in Eastern Siberia and the Far East of Russia, in particular Buryatia and the Trans-Baikal Territory. They are perfectly adapted for breeding using the herd method for keeping on steppe pastures, regardless of the season. The Buryatian native horse is a historically integral part of the local landscape, without which rational nature management and farming would be impossible. 

The Buryatian horse comes from the Mongolian breed, with which it has much in common in exterior and biological features. It also has a high level of similarity with the Trans-Baikal breed, which has also showed a high level of genetic diversity [14]. The number of breeding queens of the Buryat breed is estimated at 1000 heads, and its risk status is vulnerable [12].

The existence of the Buryatian horse almost entirely depends on natural and climatic conditions, since it is in the open air on pasture all year round. Based on the zone characteristics of herd horse breeding, the habitat of the Buryatian horses was assessed as sharply continental, cold, and moderately humid [15]. A feature of the temperature regime is large annual amplitudes that reach 70–75 degrees. 

The ecology, structure, and diversity of the microbial community of the equine gastrointestinal tract have been the subject of research in recent years [1,2,3,4,5,6,7,8]. The equine hindgut is an anaerobic fermentation ecosystem comprised of thousands of microorganisms that play a critical role in health and energy requirements [4]. Equids are hind-gut fermenters and, unlike ruminants, they can compensate for a low-quality diet by increasing consumption of lower-quality forage or bulk [16].

Differences in the digestive system, extreme environments, and eating habits of horses can lead to distinctive gut microbiota. Changes in the fecal microbiota represent an adaptation to changes in food supply and the environment.

The diversity and community structure of the fecal microbiota of pastured horses varies over a 12-month period, and this variability reflects changes in pasture nutrient composition, which in turn is influenced by climatic conditions [2]. Hindgut microbial adaptation to winter diet occurs to a greater extent in indigenous horses [17].

The bacterial microbiota of the gastrointestinal tract and feces of the Mongolian horse, as the closest relative of the Buryatian horse, was studied, and some interesting findings on the relationship between horse breeds and gut microbiota were revealed [4,5]. It was shown that the relative abundance of 31.25% (5/16) phyla and 40% (30/75) genera was significantly different between Mongolian and Thoroughbred horses living in Inner Mongolia (China) [4]. Studies on the composition of the fecal microbiota of Mongolian horses showed that, although the composition of the gut microbial community was similar to the microbiota of Thoroughbred horses at the phylum level (with the phyla *Firmicute*s and *Bacteroidetes* dominating), differences were found at the genus level, explained by environment influences [4,5]. There have been very few studies using metagenomic techniques to examine the dynamics of microbial communities in the feces of horses maintained on pasture [2,18]. 

The diversity of the fecal microbiota of the Buryatian horse has not yet been studied. We assumed that the composition of microbial communities in the feces of horses maintained on pasture would change in different seasons, depending on grass availability and different plant diets. In our study, we investigated the seasonal fluctuations in the diet composition of steppe pasture and the changes in the structure and composition of fecal microbiota in horses grazing on steppe grass pastures in the warm and cold season. The composition of the fecal microbiota of Buryatian horses was studied using the NGS method for the first time. 

## 2. Materials and Methods

### 2.1. Sample Collection

Fresh fecal samples were collected from healthy Buryatian horses grazing warm- (July, summer) and cold season (November, meteorological winter) grass pastures. The total number of horses in each group was 14 and consisted of 11 ♂ and 3 ♀, with a mean age of 5.45 years (range, 0.8 to 13 years). All horses were kept on two different farms in the Republic of Buryatia (Russia) for at least 12 months, were grazed on the steppe, and had not experienced any recent changes in housing conditions. The climatic conditions of the region are characterized by a low moisture supply, which consists of 200–350 mm of annual precipitation. The average annual air temperature minimum ranges from −40 to −45 °C, and the absolute minimum reaches −55 °C. The period with temperatures above 0 °C lasts 70–120 days, the sum of temperatures above 10 °C is 1600–1800°. The summer is warm, the average monthly temperature in July is 19.2 °C, and the highest temperatures are 36–38 °C. The mountainous relief to a large extent redistributes the incoming heat and moisture, as a result of which, even in small spaces, habitats with sharply different microclimates are formed [19]. The ambient air temperature at the time of sampling was −7 °C in November and +27 °C in July. The freshly excreted feces were collected in sterilized plastic sealing bags, stored in an icebox, and returned to the laboratory. Samples were collected directly after defecation from the subsurface fecal layer at three points. For DNA extraction, 1 mL of the freshly excreted feces were collected aseptically into fecal collection tubes for safe sample storage and transportation, using a DNA/RNA Shield (Zymo Research, Irvine, CA, USA). For microhistological fecal analyses, 20 mg of the freshly excreted feces was collected in sterilized plastic sealing bags. The samples were transported to the laboratory and were kept in a refrigerator.

### 2.2. Ethical Considerations

No procedures had to be performed on the animals included in this study. Therefore, ethical approval was not required for this research.

### 2.3. Microhistological Fecal Analysis

This technique involves preparing a small amount of fecal sample on a slide and then determining the occurrence of individual plant species across up to 100 fields on the slide. Plant species are identified by their histological characteristics, such as cell size and shape and cuticle structure [20,21]. Microhistology is known to provide good precision for diet and is a commonly used tool for many wildlife species [21]. However, it can have a low accuracy, due to differential digestion of plant species, particularly tending to underestimate the abundance of forbs, which can be more completely digested [22]. Microhistology is also labor intensive, requiring extensive training for accurate determination of plant species.

### 2.4. DNA Extraction, Amplicon Library Construction, Sequencing, and Data Analysis

Total genomic DNA from horse feces was extracted using a Quick-DNA™ Fecal/Soil Microbe Miniprep Kit (Catalog #6010) (Zymo Research, Irvine, California, USA) according to the instructions provided by the manufacturer and stored at −20 °C. The quality and concentration of DNA were assessed spectrophotometrically with a NanoDrop™8000 instrument (Thermo Fischer Scientific, Waltham, MA, USA).

16S rRNA gene amplification and MiSeq sequencing were carried as a part of the project “Large-scale search, study of microorganisms and microbial communities, farm animals and animal products” (https://microbiomeatlas.ru/, accessed on 15 May 2023), according to protocols [23,24]. The protocols showed a high convergence of results. The 16S rRNA gene fragments were amplified using PCR with the primers 341F and 806R [25]. PCR fragments were barcoded using a Nextera XT Index Kit v.2 (Illumina, San Diego, CA, USA) and sequenced on a Illumina MiSeq (2 × 300 nt paired-end reads). Overlapping reads were merged using FLASH v.1.2.11 [26]. Low-quality reads were excluded, and the remaining sequences were clustered into operational taxonomic units (OTUs) at 97% identity using the USEARCH v.11 program [27]. Chimeric sequences were removed during clustering with the USEARCH algorithm. To calculate the relative abundances of OTU, all 16S rRNA gene sequences were mapped to OTU sequences at a 97% global identity threshold using Usearch [28]. OTUs comprising only a single read were discarded. The taxonomic identification of OTUs was performed using searches against the SILVA v.138 rRNA sequence database using the VSEARCH v. 2.14.1 algorithm [28]. 16S rRNA gene fragments were amplified from DNA extracted from other parts of the samples prepared, using a two-stage PCR strategy [29] with the primers 515F [30] and Pro-mod-805R [31]. PCR of every DNA sample was carried out in duplicate; the detailed amplification protocol was described previously [23]. The libraries were checked using agarose gel and pooled equimolarly. The final pool was purified with a QIAquick Gel Extraction Kit (Qiagen, Hilden, Germany) according to the manufacturer’s protocol. Sequencing was performed with a MiSeq™ Personal Sequencing System (Illumina, San Diego, CA, USA) using 156-bp paired-end reads.

High quality read pairs were used as an input for the DADA2 pipeline [32], with the SILVA v.138 rRNA sequence database using the VSEARCH v. 2.14.1 algorithm [28]. The taxonomy of the amplified sequence variants was assigned using a naive Bayesian classifier using the Silva138 16S rRNA gene database [33]. The obtained ASV reference sequences, sample metadata, abundance, and taxonomy tables were imported into the *phyloseq* package [34], and all further operations were performed with the *phyloseq* object. Decontamination of amplicon data was performed with the decontam R package using the “prevalence” contaminant identification method with default parameters [35]. Rarefaction curve analysis was performed with the iNEXT package [36]. 

Alpha diversity metrics, including Simpson’s index of diversity, Shannon index of entropy, and the Chao1 index for the richness of bacterial genera were calculated using the *phyloseq* [34], *vegan* [37], and *microbiome* [38] R packages and the MatLab11 package (MatWorks, Inc., Beltsville, MD, USA). Pearson correlations were used to assess the associations between the microbial alpha diversity and seasonal changes. A value of *p* less than 0.03 was considered to be statistically significant. Relative abundances of microbial taxa were summarized at phylum, class, family, and genus level and included only those microbial taxa that represented >1% of the total community in at least one sample. Microbial phyla, classes, families, and genera with relative abundances <1% in all samples were grouped as “other phyla”, “other class”, “other family”, and “other genera”, respectively. Differences between microbial communities were determined using Bray–Curtis dissimilarity, which takes into account the presence or absence of a species and the relative abundance. A principal coordinate analysis (PCoA) was performed in the MatLab11 package, with the clustering of samples based on the first two principal coordinates. Preliminary processing for data standardization was carried out according to the recommendations in [39]. 

## 3. Results

### 3.1. Microhistology

A common feature of the flora of the studied steppes is the high phytocenotic significance of dense sod grasses, which are indicators of the dry steppes of Southern Siberia (*Poa botryoides*, *Agropyron cristatum*, *Koeleria cristata*) [40]. The role of pasture-resistant plants (*Potentilla acaulis*, *Artemisia frigida*, *Carex duriuscula*) is also significant [19]. The main diversity of grass vegetation in the forest–steppe landscapes of Transbaikalia is made up of mesoxerophyte grass communities, which grow in drier conditions than mesophytes but wetter than xerophytes, mainly meadow steppes with dominant *Aster alpinus*, *Bupleurum scorzonerifolium*, *Carex pediformis*, *Potentilla longifolia*, *Schizonepeta multifida,* and others [19]. The most species-rich families are characteristic of the steppe flora of Southern Siberia. The steppe flora of Buryatia contains a number of endemic plants, especially in more southern regions [40]. However, endemics are not distributed everywhere (including in the studied areas) and, moreover, do not play a significant phytocenotic role in the steppe communities, having, as a rule, a very small abundance [19]. No endemic plants were identified in our study. Out of 47 and 23 species found in the summer and winter fecal samples, accordingly (Table 1), there were an average of 18.9 per sample in summer and 10.6 per sample in winter. In winter samples, the diet was mostly made up of forbs (93%, including 51% *Schizonepeta multifida*), with 7% graminoids and 0.43% shrubs (Table 1). Other grass species found at high abundance in winter diet were *Aster alpinus* (9.4%), *Artemisia sieversiana* (8.7%), *Heteropappus altaicus* (6.8%), and *Potentilla acaulis* (6.1%). The most abundant graminoids were *Poa botryoides* (2.9%) and *Carex pediformis* (2.4%). The diet composition of horse fecal samples in summer using microhistological analysis showed a greater abundance of graminoids (41.9%), including *Poa botryoides* (14.3%), *Agropyron cristatum* (11.1%), and *Carex pediformis,* which was found in all samples and made up 8.9% of the diet. In summer, horse diets were composed of lower proportions of forbs (57.7%) and were mostly made up of *Lappula myosotis* (8.8%), *Silene repens* (8.5%), and *Aster alpinus* (7.8%). Our results showed the diet composition significantly varied for the graminoids *Poa botryoides*, *Agropyron cristatum*, and *Achnatherum splendens,* and the forbs *Schizonepeta multifida*, *Artemisia sieversiana*, *Lappula myosotis,* and *Cymbaria dahurica* in the warm and cold season (*p* ≤ 0.003 after Bonferroni correction for multiple comparisons).

### 3.2. Analysis of the Fecal Microbiota (Microbial Communities) Using 16S rRNA Profiling

#### Seasonal-Specific Differences in Diversity of the Fecal Microbial Community

Buryatian horse fecal microbial communities showed a high diversity, both in the warm and cold season (Appendix A, Appendix A). The Shannon diversity index of Buryatian horses in the cold season was 5.51, which was significantly lower than that in the warm season (5.71, *p* < 0.01). This was reflected in the trend of differences in the richness of bacterial genera (Chao1) between the two seasons (Table 2, Appendix A). We found that the Simpson’s indexes were higher in horses grazing in the cold season than in horses in the warm season. However, the level of significance after Bonferroni correction for multiple comparisons revealed non-significant differences among the seasons. 

The microbial community of both seasons comprised 28 phyla, 14 of which were present at ≥1% abundance of the total community in at least one sample, while the remaining 14 phyla (other phyla) had abundances <1%. Overall, 14 taxa were chosen for comparison at the phylum level. The microbial community was dominated by two phyla, *Firmicutes* (62.9% in winter; 49.9% in summer) and *Bacteroidetes* (25.3% in winter; 26.7% in summer), which together accounted for >80% of the overall microbial abundance (Figure 1, Table 2 and Appendix A). The fecal archaeal community was the most diverse and abundant in horses grazing warm season grass pasture.

In the summer fecal samples, *Euryarchaeota* and *Halobacterota* were present at ≥1% abundance in 4 and 5 horses, accordingly. In the winter fecal samples, *Euryarchaeota* were detected in 3 horses only, but their relative abundance was less than 1%. Other archaeal phyla were not found. Horses grazing warm season grass pasture appeared to have a higher abundance of *Cyanobacteria*, *Fibrobacterota,* and *Verrucomicrobiota* and a lower abundance of *Campylobacterota* and *Desulfobacterota* than horses grazing cold season grass pasture (Figure 1, Table 3). Among the top 14 phyla in relative abundance, *Firmicutes*, *Campylobacterota*, *Desulfobacterota,* and *Deferribacterota* were significantly enriched in the gut of horses in the cold season (Appendix A). In contrast, the relative abundance of *Verrucomicrobiota*, *Euryarchaeota*, *Fibrobacterota*, *Cyanobacteria,* and *Synergistota* in Buryatian horses was significantly higher in the warm season (Appendix A). 

At a more detailed level, several taxa were significantly less abundant in samples collected in the summer than samples collected in the winter. These were assigned to the bacterial families *Lactobacillaceae*, *Muribaculaceae*, *Erysipelotrichaceae*, *Helicobacteraceae*, *Desulfovibrionaceae*, *Peptostreptococcaceae,* and *Bacteroidaceae* (Table 3). Other taxa were more abundant in samples collected in the summer than in samples collected in the winter, belonging to the bacterial families *Acidaminococcaceae*, *Rikenellaceae*, UCG-010, p-251-o5, F082, WCHB1-41, and unidentified taxa of Order_*Gastranaerophilales*. 

*Tannerellaceae*, *Helicobacteraceae*, and *Bifidobacteriaceae* were not detected in horse guts in summer. Horses grazing in winter lacked WCHB1-41, p-251-o5, and some *Bacteroidales* families, including BS11_gut_group and UCG-001. Among the top 30 families, the relative abundances of *Acidaminococcaceae*, *Anaerovoracaceae*, *Bacteroidaceae*, *Bacteroidales* RF16 group, *Butyricicoccaceae*, *Desulfovibrionaceae*, *Erysipelotrichaceae*, F082, *Fibrobacteraceae*, *Lactobacillaceae*, *Methanobacteriaceae*, *Muribaculaceae*, *Clostridia* UCG-014, *Gastranaerophilales*, RF39, *Peptostreptococcaceae*, *Prevotellaceae*, *Rikenellaceae*, *Ruminococcaceae*, *Saccharimonadaceae*, *Selenomonadaceae,* and UCG-010 were significantly different between the two groups (*p* ≤ 0.003) (Appendix A).

There were significant effects of season on the beta diversity of microbial families, which clustered by warm and cold season in the principal coordinate analysis (PCoA), with 45% of the variation explained by two principal coordinates (34.9% on PCs 1 and 10.1% on PCs 2) (Figure 2). This clustering by season was further reinforced by the significant differences observed in the relative abundances of microbial families and microbial genera (Appendix A).

Among the top 14 genera (Figure 3), the relative abundances of *Christensenellaceae*_R-7_group, *Clostridia*_UCG-014, *Lachnospiraceae*_AC2044*_group*, *Ligilactobacillus*, *NK4A214*_group, *Rikenellaceae*_RC9_gut_group, UCG-002, and *Ruminococcus* were significantly different between the two groups (*p* ≤ 0.003). The relative abundances of *Clostridia*_UCG-014, *Ligilactobacillus*, and *Ruminococcus* in the gut of Buryatian horses were significantly higher in the cold season, while the relative abundances of *Christensenellaceae*_R-7_group, *NK4A214*_group, UCG-002, *Lachnospiraceae*_AC2044_group, and *Rikenellaceae*_RC9_gut_ group were significantly higher in the warm season (Appendix A).

## 4. Discussion

The Buryatian horse is an important breed in Buryatia and the Trans-Baikal Territory (Russia) and is actively used in agriculture today [13]. As an indigenous breed, they have a rich genomic diversity [14] and a variety of excellent traits, such as adaptability, cold resistance, roughage resistance, disease resistance, and good stamina. The harsh climate of this herd horse breeding zone has led to the formation of a distinctive vegetation cover of pastures, which is represented by the presence of sedges, and meadow and subalpine forbs. To the south, closer to lakes and floodplains, forest–tundra vegetation is replaced by forest–steppe and steppe. On the pastures used by the Buryatian horses, 86 species of herbs and several types of shrubs have been identified [15]. 

Most studies of feral horse diets have used microhistological analyses of fecal matter [21]. We conducted a quantitative synthesis of microhistological fecal studies of the year-round grazing Buryatian horses. Diet composition significantly varied for graminoids and forbs in the warm and cold season (*p* ≤ 0.03). For wild horses on the western rangelands of North America, the seasonal diet composition consisted of graminoids (77–89%), forbs (4–15%), and browse (3–10%) [21]. Unlike these data, our results indicated that Buryatian horse diets are primarily composed of forbs (93% in the cold season and 57.7% in the warm season). There are many reasons for this difference between the two studies, including regional and climate habitat features, specific shifts in the grass community, and the response to winter snow that limited graminoid accessibility, leading to higher forbs composition. Our results showed the diet composition significantly varied for graminoids and forbs in the warm and cold seasons. Steppe graminoids, *Poa botryoides*, *Agropyron cristatum,* and different *Carex* sp., make up a significant part of the horses’ diet in the warm season. All of these species are resistant to trampling [19]. While animals prefer to eat certain species at the beginning of summer, when the leaves are tender, others can also be used in winter. *Agropyron cristatum* is a perennial grass that is considered a good fattening feed and provides valuable food from early spring to late autumn. Forbs play an important role throughout the year. The importance of wormwood *Artemisia sieversiana* increases in horse nutrition in winter. Wormwood has a bitterness and becomes suitable for forage in late autumn after frost. In winter, *Potentilla acaulis*, which hibernates with a green rosette of leaves and well-formed buds, is maintained in the diet of horses and considered a good fattening food. The importance of *Schizonepeta multifida* and *Heteropappus altaicus,* with strong frost resistance, significantly increases in the winter diet. *Schizonepeta multifida* is both a medicinal and pasture plant [41]. *Schizonepeta multifida* (or *tenuifolia*), also known in China as Jing Jie, belongs to the family *Lamiaceae* and is a perennial herbaceous plant and an herbal medicine that has been widely used for thousands of years in China, Japan, and Korea [42]. As well as the wormwood *Artemisia sieversiana*, *Schizonepeta multifida* has clear antimicrobial, antioxidant, immunomodulatory, and anti-inflammatory effects [41,42,43] and is used as medicinal plants in the complex therapy of endogenous infections [44]. Diet is the main factor that contributes to gut microbiome modification in horses [45]. Therefore, the abundance of *Schizonepeta multifida* and *Artemisia sieversiana* with high pharmacological effects in the winter diet of the Buryatian horse may be an important factor influencing gut microbiome composition and horse health. The variation in diet composition and the gut microbiota changes observed in the current study may have implications for grazing management and the preparation of conserved forages for horses susceptible to perturbations in hindgut microbial community. 

Changes in the composition and diversity of the horse gut microbial communities can reflect feeding patterns and seasonal variations in diet composition, especially in year-round grazing animals [5]. Variation sin pasture composition and fluctuations in the diversity of fecal microbiota were observed for mature adult horses maintained on pasture in New Zealand [2]. *Firmicutes* and *Bacteroidetes* were considered important when monitoring changes in microbial populations, because major shifts in bacterial diversity and abundance were found in these two phyla [2]. *Firmicutes* dominated the gut community when the pasture was growing, whereas when the pasture was drought-stressed, the abundance of *Bacteroidetes* increased. Similarly, *Firmicutes* (62.9% in winter; 49.9% in summer) and *Bacteroidetes* (25.3% in winter; 26.7% in summer) were the main dominant phyla in our study. *Actinobacteriota*, *Campylobacterota*, *Desulfobacterota*, *Spirochaetota,* and *Proteobacteria* were other dominant phyla in winter. *Verrucomicrobiota*, *Proteobacteria*, *Cyanobacteria*, *Spirochaetota*, *Halobacterota*, *Actinobacteriota,* and *Fibrobacterota* had a high relative abundance in summer. 

The bacterial microbiota of the gastrointestinal tract and feces of the Mongolian horse, as the closest relative of the Buryatian horse, have been thoroughly explored in studying the gut microbiota differences between Thoroughbred and Mongolian horses [4,5]. Studies on the composition of the fecal microbiota of Mongolian horses grazed on the prairie in September showed *Firmicutes*, *Bacteroidetes*, *Spirochaetes*, *Proteobacteria*, *Verrucomicrobia*, and *Fibrobacteres* were the predominant phyla [4]. The same dominant phyla, accounting for 97.6%, with the addition of *Kiritimatiellaeota* as another dominant were detected by Wen et al. [5]. Although the composition of the community was similar at the phylum level, the proportion of each phylum was different, except for *Firmicutes* (56%), which was the most dominant phylum in the gut of Mongolian horses in both studies. The proportion of *Bacteroidetes* varied from 16% to 33% in Wen’s and Zhao’s studies, accordingly. The proportion of *Proteobacteria* was 19% for Mongolian horses in Wen’s study, while it was 1% in Zhao’s study. 

At the family level, horses grazing cold season grass pasture had larger numbers of two lactic acid bacteria: *Lachnospiraceae* and *Lactobacillaceae*, as well as *Prevotellaceae*. Previous findings showed that increased relative abundance of lactic acid bacteria and excessive lactate production are associated with large intestinal illness in horses [3]. In the current study, we investigated the gut microbiota composition in healthy Buryatian horses, but an increased relative abundance of the lactic acid bacteria *Lachnospiraceae* was revealed in horses, both in the warm and cold seasons. It is not clear whether this is related to the breed of Buryatian horses, which are perfectly adapted for breeding using the herd method and for keeping on steppe pastures, regardless of the season. However, at the same time, the relative abundance of *Escherichia* and *Streptococcus*, which are other markers of horse intestinal diseases [3], were 0.2 and 0.1%, accordingly, in both seasons. *Prevotellaceae* has enzymes capable of fermenting and utilizing complex polysaccharides [46]. Members of the underexplored family *Muribaculaceae* were significantly enriched in the horses’ gut in winter but were barely identified in the warm season. *Muribaculaceae,* as well as members of the *Lachnospiraceae*, *Rikenellaceae*, and *Bacteroidaceae* families, were commensals, which use mucin-derived sugars and impede pathogen colonization [47]. It would be of interest to determine if these taxonomic characteristics have any impact on improving the efficiency of energy and nitrogen utilization in horses, to survive the nutritional stress in severe cold season conditions. 

The fecal archaeal community was the most diverse and abundant in horses grazing warm season grass pasture. The dominant archaeal taxa in feces were related to g. *Methanocorpusculum* and g. *Methanobrevibacter*. As demonstrated in other studies [48,49,50], methanogenic archaea are often abundant in healthy equine colons. A comparative analysis of the archaeal diversity in horse gut showed that methanogens affiliated to *Methanobacteriales* and *Methanomicrobiales* were predominant [50]. The abundance of *Methanobrevibacter* (*Methanobacteriales*) is considered a biomarker for monitoring horse intestinal health, because *Methanobrevibacter* decreases significantly in horses with large and small intestinal colic compared to healthy horses [3]. In the current study, the dominant archaeal taxa of feces were related to *Methanocorpusculum* in the warm season, while *Methanobrevibacter* were found in samples from both seasons, but its abundance was reduced in winter.

## 5. Conclusions

The first attempt to assess the diet composition and taxonomic diversity of fecal microbial communities in healthy Buryatian horses revealed significant differences in microbial community composition among the warm and cold seasons. These findings can serve as an experimental substantiation for further study of the impact of the pasture grasses *Schizonepeta multifida* and *Artemisia sieversiana*, which have antimicrobial, antioxidant, immunomodulatory, and anti-inflammatory effects, on the diversity of the gut microbiome and health of horses. The obtained results can improve management treatment protocols for horses suffering from health issues associated with nutrition or feeding.

## Figures and Tables

**Figure 1 microorganisms-11-01947-f001:**
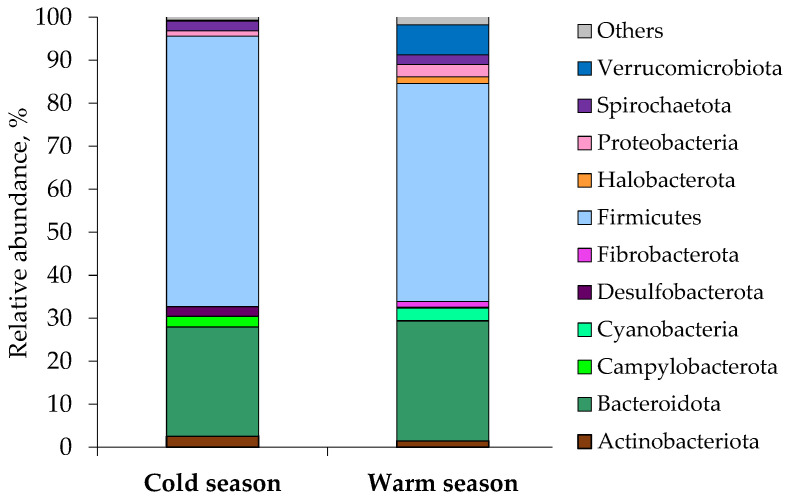
Mean relative abundance of microbial phyla identified in the fecal samples of Buryatian horses grazing cold and warm season grass pastures. “Others” included *Euryarchaeota*, *Deferribacterota* and *Synergistota*.

**Figure 2 microorganisms-11-01947-f002:**
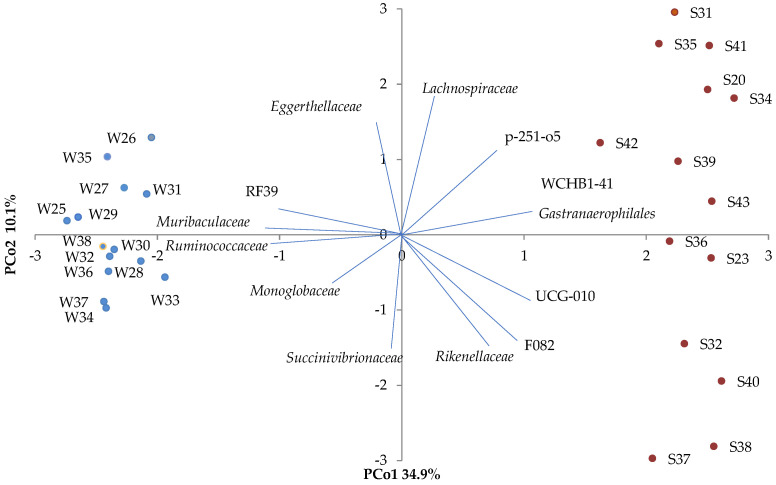
PCoA of the beta diversity of fecal microbial community in horses illustrated by season, showing the principal coordinate analysis (PCoA) of the beta diversity in the microbial community; with each fecal sample represented as a dot (i.e., each dot represents a horse within a season: W—cold season, S—warm season). The principal coordinates explain 34.9% (PCo1) and 10.1% (PCo2) of the variation.

**Figure 3 microorganisms-11-01947-f003:**
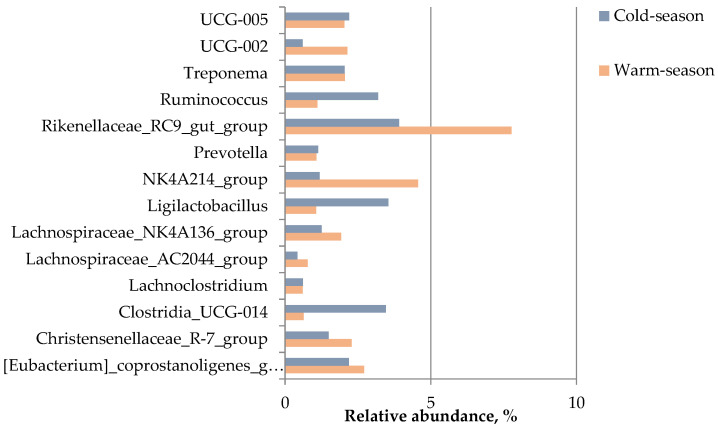
Top 14 genera in abundance.

**Table 1 microorganisms-11-01947-t001:** Diet composition of Buryatian horse fecal samples from microhistological analyses.

Summer		Winter	
Species	Mean%	Species	Mean%
Total graminoids:	41.9 ± 11.7	Total graminoids:	7 ± 5.3
*Poa botryoides*	14.3	*Poa botryoides*	2.9
*Agropyron cristatum*	11.1	*Carex pediformis*	2.4
*Carex pediformis*	8.9	*Agropyron cristatum*	1.2
*Stipa capillata* L.	3.0	*Koeleria cristata*	0.3
*Achnatherum splendens*	1.7	*Achnatherum splendens*	0.2
*Koeleria cristata*	1.4		
*Carex* sp.	1.1		
*Carex duriuscula* C. A. Meyer 1831	0.8		
Total forbs:	57.7 ± 11.8	Total forbs:	92.6 ± 5.9
*Lappula myosotis*	8.9	*Schizonepeta multifida*	51.43
*Silene repens*	8.5	*Aster alpinus*	9.4
*Aster alpinus*	7.8	*Artemisia sieversiana*	8.7
Unknown grass 6	4.3	*Heteropappus altaicus*	6.8
*Potentilla acaulis*	3.9	*Potentilla acaulis*	6.1
*Allium altaicum*	2.6	*Heteropappus biennis*	2.5
*Thalictrum etaloideum* L.	2.5	*Lappula myosotis*	2.4
Unknown grass 2	2.3	*Cymbaria dahurica*	1.4
*Heteropappus biennis*	2.3		
*Serratula centauroides L. s. str.*	1.7		
*Achnatherum splendens*	1.7		
*Pulsatilla turczaninovii*	1.5		
*Artemisia dracunculus* L.	1.3		
*Saussurea salicifolia* L.	1.1		
*Phlomis tuberosa* L.	1.0		
*Nonea rossica*	1.0		
*Potentilla longifolia*	0.8		
*Schizonepeta multifida*	0.8		
*Artemisia frigida* Wil	0.7		
*Axyris amaranthoides* L.	0.6		
*Leontopodium leontopodoides*	0.4		
Unknown grass 9	0.4		
*Hypecoum (Chiazospermum) erectum*	0.3		
Unknown grass 4	0.3		
Unknown grass 5	0.3		
*Astragalus adsurgens* Pallas	0.3		
*Artemisia gmelinii* Webb ex Stechm.	0.3		
*Descurainia sophia* (L.) Webb. et Berth.	0.2		
*Potentilla bifurca*	0.2		
*Allium anisopodium* Ledeb	0.2		
*Pulsatilla multifida*	0.2		
*Cymbaria dahurica*	0.2		
*Heteropappus altaicus*	0.1		
*Artemisia sieversiana*	0.1		
*Saposhnikovia divaricata* (Turcz.) Hiroe	0.1		
*Thalictrum squarrosum* Stephan ex Willd.	0.1		
*Galium verum* L.	0.1		
*Leonurus* sp.	0.1		
Total shrubs:	0.4 ± 1.3	Total shrubs:	0.43 ± 1.3
*Kochia prostrata* Sschraber	0.4	*Kochia prostrata Sschraber*	0.43
*Spiraea* sp.	0.1		

**Table 2 microorganisms-11-01947-t002:** Analysis of alpha diversity at 97% similarity.

Season	ASVs/OTUs	Chao1	Shannon Index	Simpson Index
Warm season	887	1263.7 ± 203	5.74 ± 0.41	0.986 ± 0.024
Cold season	114	480.8 ± 35.7	5.51 ± 0.11	0.991 ± 0.003
*p*-value *	<0.001	<0.001	<0.001	0.449

* Level of significance was *p* ≤ 0.05.

**Table 3 microorganisms-11-01947-t003:** Phyla, classes, and families with relative abundance of >1% in the Buryatian horse fecal microbiota.

Phylum	Class	Family	Cold Season Mean Relative Abundance, %	Warm Season Mean Relative Abundance, %
Firmicutes			62.95	50.74
	Clostridia		40.64	42.60
		Lachnospiraceae	11.4	13.1
		Oscillospiraceae	7.7	9.6
		Ruminococcaceae	9.4	3.2
		Christensenellaceae	1.5	2.3
		Order_Clostridia UCG-014	3.5	
		Peptostreptococcaceae	1.8	
		UCG-010		3.9
		Alkalibacteraceae		2.1
		Anaerovoracaceae		1.8
		Hungateiclostridiaceae		1.2
	Bacilli		20.15	5.87
		Erysipelatoclostridiaceae	3.2	1.2
		Lactobacillaceae	11.5	1.9
		Planococcaceae		1.1
	Negativicutes		1.12	2.20
		Acidaminococcaceae		1.9
Bacteroidota			25.33	27.90
	Bacteroidia		26.20	27.90
		Prevotellaceae	10.9	6.7
		Rikenellaceae	4.5	9.1
		Muribaculaceae	6.4	
		Bacteroidaceae	1.6	
		p-251-o5		4.4
		F082		3.5
		Bacteroidales_UCG-001		1.8
		Bacteroidales_RF16_group		1.3
Actinobacteriota			2.61	1.45
	Coriobacteriia		2.18	1.15
		Eggerthellaceae	1.1	
Campylobacterota			2.50	
	Campylobacteria		2.55	
		Helicobacteraceae	2.3	
Cyanobacteria				2.86
	Vampirivibrionia			2.86
		Gastranaerophilales		2.9
Desulfobacterota			2.23	
	Desulfovibrionia		2.25	
		Desulfovibrionaceae	2.2	
Fibrobacterota				1.34
	Fibrobacteria			1.34
		Fibrobacteraceae		1.3
Halobacterota				1.56
	Methanomicrobia			1.49
		Methanocorpusculaceae		1.5
Proteobacteria			1.26	2.88
	Alphaproteobacteria			1.98
		Caulobacteraceae		1.1
Spirochaetota			2.23	2.20
	Spirochaetia		2.16	2.12
		Spirochaetaceae	2.2	2.1
Verrucomicrobiota				7.00
	Kiritimatiellae			6.49
		WCHB1-41		6.5

## Data Availability

FASTQ sequences of this metagenomic sample have been deposited in the NCBI Short Read Archive under BioProject PRJNA987555.

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
