# Peer review of "Fecal Microbiota and Diet Composition of Buryatian Horses Grazing Warm- and Cold-Season Grass Pastures"

_microorganisms, 2023, doi:10.3390/microorganisms11081947_

Round 1

Reviewer 1 Report

1. The article is impeccably written, presenting a comprehensive and insightful subject matter analysis. The author shall add one more section regarding “how pasture will be changed in different seasons depending on grass availability and different plant diets” mechanisms of action or wherever needed.

2. The author can also add some more references as it increases the quality of the article.

3. Please check the formatting in the references section as in line no. 388 the DOI format is different from the format of the journal please cross-check all the references it should be in the same format provided by the Journal.

Moderate editing of english is required

Author Response

Author’s Reply to the Review Report

We are very grateful to the Reviewer for being so kind to our study.

  1. The article is impeccably written, presenting a comprehensive and insightful subject matter analysis. The author shall add one more section regarding “how pasture will be changed in different seasons depending on grass availability and different plant diets”mechanisms of action or wherever needed.

Response 1: We have added some conclusions concerning how pasture changed in warm and cold seasons and what grasses were available. In line 323-336: Our results showed diet composition significantly varied for graminoids and forbs in warm- and cold-season. Steppe graminoids: Poa botryoides, Agropyron cristatum and different Carex sp., make up a significant part of the horse’s diet in warm season. All of these species are resistant to trampling. Some of them animals prefer to eat at the beginning of summer, while the leaves are tender, others can be used in winter also. Agropyron cristatum is a perennial grass that is considered good fattening feed, provides valuable food from early spring to late autumn. Forbs play an important role throughout the year. The importance of various species of wormwood, mainly Artemisia sieversiana, increases in the horse nutrition in winter. Wormwood has bitterness and become suitable for forage in late autumn after frost. In winter, Potentilla acaulis, which hibernates with a green rosette of leaves and well-formed buds, is preserved in the diet of horses and considered to be a good fattening food. The importance of Schizonepeta multifida and Heteropappus altaicus with strong frost resistance significantly increases in the winter diet. Schizonepeta multifida is both medicinal and pasture plant [41].

  1. The author can also add some more references as it increases the quality of the article.

Response 2: We supplemented the list of references with 10 more references.

  1. Please check the formatting in the references section as in line no. 388 the DOI format is different from the format of the journal please cross-check all the references it should be in the same format provided by the Journal.

Response 3: Done

Reviewer 2 Report

Dear authors

Thanks for your effort and nice presentation.

However, some points should be considered during your revision process.

1. In the abstract section, the conclusion statement should be added at the end lines.

2. The conclusion section is a repetition of the results.  Some recommendations or advice should be included.

3. In the comparison with the introduction section, the discussion section should be more in-depth and supported by more references.

4. What are the benefits or the impact of the obtained results on the Veterinary field of equine?.  This point should be highlighted in the manuscript.

Best wishes

Author Response

Author’s Reply to the Review Report

We are very grateful to the Reviewer for being so kind to our study.

  1. In the abstract section, the conclusion statement should be added at the end lines.

Response 1: Done

  1. The conclusion section is a repetition of the results.  Some recommendations or advice should be included.

Response 2: Done. We have included some recommendations in the conclusion section.

  1. In the comparison with the introduction section, the discussion section should be more in-depth and supported by more references.

Response 3: Done. We have expanded the discussion section and supplemented the list of references with 10 more references.

  1. What are the benefits or the impact of the obtained results on the Veterinary field of equine?.  This point should be highlighted in the manuscript.

Response 4: In line 339: As well as wormwood Artemisia sieversiana, Schizonepeta multifida has a clear antimicrobial, antioxidant, immunomodulatory and anti-inflammatory effect [41-43] and is used as medicinal plants in the complex therapy of endogenous infections [44]. The diet is the main factor that contributes to gut microbiome modification in horses [45]. Therefore, the abundance of Schizonepeta multifida and Artemisia sieversiana with high pharmacological effects in the winter diet of the Buryatian horse may be an important factor influencing the gut microbiome composition and horse health.

In line 403: The findings can serve as an experimental substantiation for further study of the impact of pasture grasses Schizonepeta multifida and Artemisia sieversiana, which have an antimicrobial, antioxidant, immunomodulatory and anti-inflammatory effect, on the diversity of the gut microbiome and health of the horse. The obtained results can improve management treatment protocols for horses suffering from health issues associated with nutrition or feeding.

Reviewer 3 Report

The manuscript presented by the authors provides a comprehensive study on the “Fecal Microbiota and Diet Composition of Buryatian Horses Grazing Warm- and Cold-Season Grass Pastures”, using a large and valuable dataset. The study is important in deepening our understanding of “fecal microbiota in horse”. However, there are some issues for improvement in the manuscript.

Major comments:

1.  In Figure 1, (a) and (b) in the figure should be replaced by Cold and Warm, and add the description in the legend. Additionally, alpha diversity, rarefaction curve, sample composition should to show overview the microbiota, like Figure 1A/B/C/E in EasyAmplicon (https://doi.org/10.1002/imt2.83).

2.  The figures need to be improved. Such as the webserver ImageGP (https://doi.org/10.1002/imt2.5) can generate high quality figures and with reproducible scripts.

3.  In figure 2, too many bacterial names in the figure, please only list the top 10 in order to read.

4.  In figure 3, the top 14 genus? You should overview the different genus using Manhattan plot, then plot the overall different in heatmap, and finally plot the top N different genus in boxplot, such as Figure 1H, Figure 2E and Figure 1A in EasyAmplicon.

Author Response

We are very grateful to the Reviewer for being so kind to our study.

  1. In Figure 1, (a) and (b) in the figure should be replaced by Cold and Warm, and add the description in the legend. Additionally, alpha diversity, rarefaction curve, sample composition should to show overview the microbiota, like Figure 1A/B/C/E in EasyAmplicon (https://doi.org/10.1002/imt2.83).

Response 1: Done. Additionally, alpha rarefaction curve (Figure S2) and heatmap of species abundance clustering at genus level (Figure S3) were added.

2.  The figures need to be improved. Such as the webserver ImageGP (https://doi.org/10.1002/imt2.5) can generate high quality figures and with reproducible scripts.

Response 2: Done. All figures have been improved.

3. In figure 2, too many bacterial names in the figure, please only list the top 10 in order to read.

Response 3: Done, Figure 2 have been improved.

4. In figure 3, the top 14 genus? You should overview the different genus using Manhattan plot, then plot the overall different in heatmap, and finally plot the top N different genus in boxplot, such as Figure 1H, Figure 2E and Figure 1A in EasyAmplicon.

Response 4: Done, Figure 3 have been re-made.

Reviewer 4 Report

The research is interesting however some information should be added in the paper as well as improvements mainly in the discussions

Summary

There is no conclusion in the summary, only result. After changing the conclusion of the work, it must be added in the summary.

Keywords should not be the same as the title

Based on advances in animal nutrition, it is known that the season is one of the factors that cause changes in the composition of animal diets. studies are not recent.

The work does not present innovation.

Methods

In research that uses animals, it is necessary to add the authorization protocol of the ethics committee for the use of animals. What is not included in this research.

Where are the properties located; What are the climatic conditions of the region observed during the collection period; What is the average temperature of the hot and cold seasons mentioned in the text;  Were the feces collected on every day of July and November; How many collections were performed per animal? What is the average amount of feces collected.

It is suggested to add the chemical composition of the pasture in its different seasons, as well as its composition of the pasture.

Discussion

Revision and improvement are needed. The sentences are general.

329-330 – It is recommended that this discussion be focused on and compared to bovines and yes animals of the same species of your research. As well as similar diets, in your discussion in addition to comparing cattle, these are fed high grain diets.

Conclusions

Conclusions should be straight to succinct based on your results. The way it is mainly described in the first and third paragraphs are results and not conclusions.

Author Response

We are very grateful to the Reviewer for being so kind to our study.

Summary

There is no conclusion in the summary, only result. After changing the conclusion of the work, it must be added in the summary.

       Response 1: Done. We have changed the conclusion and added this changing in the summary.

Keywords should not be the same as the title

Response 2: Done. In line 26 we changed key words. Keywords: microbial diversity; 16S rRNA; horse indigenous breed

Based on advances in animal nutrition, it is known that the season is one of the factors that cause changes in the composition of animal diets. studies are not recent.

The work does not present innovation.

       Response 3: Indeed, the season is one of the factors that cause changes in the composition of animal diets. But we investigated the seasonal fluctuations in the diet composition of Buryatian horse using microhistological fecal analysis for the first time. In addition, the diversity of the faecal microbiota of the Buryatian horse has not been studied yet.

Methods

In research that uses animals, it is necessary to add the authorization protocol of the ethics committee for the use of animals. What is not included in this research.

Response 4: We added in line 121: Ethical Considerations. No procedures had to be performed on the animals included in this study. Therefore, ethical approval was not required for this research.

Where are the properties located; What are the climatic conditions of the region observed during the collection period; What is the average temperature of the hot and cold seasons mentioned in the text;  Were the feces collected on every day of July and November; How many collections were performed per animal? What is the average amount of feces collected.

Response 5: We added in line 104: The climatic conditions of the region are characterized by low moisture supply, which consist of 200–350 mm of annual precipitation. The average of the annual air temperature minimums ranges from -40 to -45 °C, the absolute minimum reaches -55 °C. The period with temperatures above 0 °C lasts 70–120 days, the sum of temperatures above 10 °C is 1600–1800°. The summer is warm, the average monthly temperature in July is 19.2 °C, and the highest temperatures are 36–38 ° C. The mountainous relief redistributes to a large extent the incoming heat and moisture, as a result of which, even in small spaces, habitats with a sharply different microclimate are formed [19]. The ambient air temperature at the time of sampling was −7 °C in November and +27 °C in July.

In line 116: For DNA extraction 1 ml of the freshly excreted feces were collected aseptically into fecal collection tubes for safe sample storage and transportation DNA/RNA Shield (Zymo Research, USA). For microhistological fecal analyses 20 mg of the freshly excreted feces were collected in sterilized plastic sealing bags.

It is suggested to add the chemical composition of the pasture in its different seasons, as well as its composition of the pasture.

 Response 6: Unfortunately, we haven’t conducted the analyses of chemical composition of the pasture in current study. But this important aspect will be studied in the future research.

Discussion

Revision and improvement are needed. The sentences are general.

 Response 7: Done. We have expanded the discussion section.

329-330 – It is recommended that this discussion be focused on and compared to bovines and yes animals of the same species of your research. As well as similar diets, in your discussion in addition to comparing cattle, these are fed high grain diets.

 Response 8: We discuss our result in comparison with the previous studies on the faecal microbiota of Mongolian horses, mainly.  The Mongolian horse is the closest relative of the Buryatian horse and graze on the prairie under similar climatic conditions. In addition, we discussed our results in the context of horse health and previously identified microbiome patterns in horses with intestinal diseases. The results obtained for cattle were used for demonstration potential metabolic activity of Prevotellaceae, only.

Conclusions

Conclusions should be straight to succinct based on your results. The way it is mainly described in the first and third paragraphs are results and not conclusions.

Response 9: We have re-written the conclusion section. In lines 410-416: The findings can serve as an experimental substantiation for further study of the impact of pasture grasses Schizonepeta multifida and Artemisia sieversiana, which have an antimicrobial, antioxidant, immunomodulatory and anti-inflammatory effect, on the diversity of the gut microbiome and health of the horse. The obtained results can improve management treatment protocols for horses suffering from health issues associated with nutrition or feeding.

Reviewer 5 Report

Dear authors, the work needs a little finishing, I offer you an option with my marks, which would make the article clearer. Please rearrange the figures, figure 2 is especially confusing.

Author Response

Author’s Reply to the Review Report

We are very grateful to the Reviewer for being so kind to our study and useful advices.

  1. Although there were few females, were they lactating? Did it affect the composition of the microbiome?

Response 1: Indeed, some females were lactating. But we didn’t reveal features of microbiome composition in lactating horses in the current study. Moreover, no effect of gender was found on the faecal microbiota.

  1. It is not clear why it was necessary to use a mixture of two different gene fragments? for what reason and on the basis of which fragment the taxonomy was nevertheless determined, this is important .

Response 2: The study was carried out as part of a project in which several institutions participated, it turned out that the extracted DNA from half of the native samples was sent to the Kurchatov Center for Genome Research for amplification and sequencing. The extracted DNA from other half of the native samples was sent to Institute of Bioengineering for amplification and sequencing. Amplification and sequencing were performed according methods used in these laboratories. This explains the use of different primers and programs.

  1. please add a bit about the pasture botanical landscape, degree of endemism?

Response 3: Done. In lines 189-202: A common feature of the flora of the studied steppes is the high phytocenotic significance of densely sod grasses, which are indicators of the dry steppes of Southern Siberia (Poa botryoides, Agropyron cristatum, Koeleria cristata) [40]. The role of pasture-resistant plants (Potentilla acaulis, Artemisia frigida, Carex duriuscula) is also high [19]. The main diversity of grass vegetation in the forest-steppe landscapes of Transbaikalia is made up of mesoxerophyte grass communities, which growing in drier conditions than mesophytes, but wetter than xerophytes, mainly meadow steppes with dominant Aster alpinus, Bupleurum scorzonerifolium, Carex pediformis, Potentilla longifolia, Schizonepeta multifida and others [19]. The most species-rich families are characteristic of the steppe flora of Southern Siberia. The steppe flora of Buryatia contains a number of endemic plants, especially in more southern regions [40]. However, endemics are not distributed everywhere (including in the studied areas) and, moreover, do not play a significant phytocenotic role in the steppe communities, meeting, as a rule, with a very small abundance [19]. No endemic plants were identified in our studies.

  1. usually two figures are given: dominant and minor phyla ...It looks strange that Halobacteriota are not visible, but are indicated in the legend, while Others are visible, but what is it? Uncultivated?

If possible, please provide this information in more detail.

Response 4: Done. Figure 1 has been improved. Halobacteriota were found in horses grazing warm-season only. It’s relative abundance was 2% in gut microbiom. “Others” included Euryarchaeota, Deferribacterota and Synergistota, which were present at ≥ 1% abundance of the total community, in at least one sample.

  1. At PCoA plot what are red captions mean? Maybe author miss vectors there? Figure need to be clarify.

Response 5: Done, Figure 2 has been improved.

  1. Figure 3. is too big. Plot could be rotated clockwise so the axis with genera aligned vertically. For example https://rstudio-pubs-static.s3.amazonaws.com/4305_8df3611f69fa48c2ba6bbca9a8367895.html

Response 6: Done, Figure 3 has been improved

Round 2

Reviewer 2 Report

Thanks for your revision.

Reviewer 3 Report

All the data visulization need to be improved. However, no quailty improved. Far away from the publication standard.

Reviewer 4 Report

Favorable the changes made by the authors